# CO$_2$ Capture of the Gas Emission, Using a Catalytic Converter and Airlift Bioreactors with the Microalga *Scenedesmus dimorphus*

**Citlalli Adelaida Arroyo [1], José Luis Contreras [1,*], Beatriz Zeifert [2] and Clementina Ramírez C. [1]**

[1] Departamento de Energía, Universidad Autónoma Metropolitana-Azcapotzalco, CBI, Av. Sn. Pablo 180, Mexico City, C.P. 02200, Mexico

[2] Instituto Politécnico Nacional, ESIQIE, UPALP, Zacatenco, Ciudad de Mexico C.P. 02200, Mexico

\* Correspondence: jlcl@correo.azc.uam.mx; Tel.: +52-55-5591911047

**Abstract:** A process composed by a catalytic converter and three sequential Airlift photobioreactors containing the microalga *Scenedesmus dimorphus* was studied to capture CO$_2$, NOx, and CO from emissions of a steam boiler which was burning diesel. The catalytic converter transformed to CO$_2$ a maximum of 78% of the CO present in the combustion gas. The effects of shear rate, light intensity, and light/dark cycles on the biomass growth of the algae were studied. It was observed that at low shear rates (Re $\approx$ 3200), a high productivity of 0.29 g$_{cel}$ L$^{-1}$ d$^{-1}$ was obtained. When the microalga was exposed to 60.75 $\mu$mol·m$^{-2}$·s$^{-1}$ of intensity of light and a light/dark cycle of 16/8 h, a maximum productivity of 0.44 g$_{cel}$ L$^{-1}$ d$^{-1}$ and a maximum CO$_2$ fixation rate 0.8 g CO$_2$ L$^{-1}$·d$^{-1}$ were obtained. The maximum CO$_2$ removal efficiency was 64.3%, and *KLa* for CO$_2$ and O$_2$ were 1.2 h$^{-1}$ and 3.71 h$^{-1}$ respectively.

**Keywords:** CO$_2$ capture; *Scenedesmus dimorphus*; microalgae; airlift-bioreactor; CO conversion; catalytic-converter

## 1. Introduction

Electricity generation in Mexico has proven to be a significant source of air pollution at the national level [1]. In 2015, the electricity sector produced 124.85 million of tons of CO$_2$, 205,787 tons of NOx and 60,968 tons of CO [2,3]. In general, steam boilers produce pollutant gases such as CO, NO, NO$_2$, hydrocarbons, SO$_2$, and solid particles or soot. Given the need to regulate the emissions of these gases, there is a wide variety of post-combustion treatments among which are the chemical absorption and biological fixation processes using microalgae [4–6].

One of the limitations for the use of CO and NO emissions in the cultivation of microalgae is its low solubility of 0.0068 and 0.0026 g·L$^{-1}$, respectively [7]. That is why they must first be transformed into more soluble compounds, such as CO$_2$ and NO$_2$ (0.0386 mol·L$^{-1}$·atm$^{-1}$ and more than 1 mol·L$^{-1}$ in water respectively). The CO and NO can be burned catalytically to CO$_2$ and NO$_2$, using a catalytic converter with noble metals or metal oxides as catalysts. These converters are composed of ceramic materials in the form of a honeycomb coated with alumina where metals like platinum are supported as active sites and other oxides such as CeO$_2$, TiO$_2$, ZrO$_2$, SiO$_2$-Al$_2$O$_3$ [8]. These converters can use the thermic energy of combustion gases to carry out the oxidation reactions [9].

Emissions from steam generators also contain volatile organic compounds (VOCs) such as polycyclic aromatic hydrocarbons, carcinogenic aldehydes or carbon in the form of soot. The conversion of unburned hydrocarbons to CO$_2$ is also carried out by the catalysts mentioned above. The presence of sulfur compounds such as SO$_2$ may also be present and depends on the quality of



the fuel. In general, the diesel currently sold in Mexico City does not contain significant amounts of sulfur (< 1 vol.%), so it did not represent a problem in this study. To remove the soot formed by unburnt carbon particles, zeolite-based filters with Pt are used to retain and convert the particles to $CO_2$ [8]. Sequentially, it is necessary to place a catalytic converter for the treatment of diesel combustion emissions. In our previous study, we have observed that the removal of $NO_2$ can be carried out in microalgae bioreactors as part of its metabolism; however, this was not the case for $SO_2$ since this gas decreases its growth at a concentration of 60 ppm [10].

As is known, there are two types of bioreactors for cultivating algae to sequester $CO_2$ that include open raceway ponds and closed bioreactors [11,12]. The closed bioreactors allow better control of the operating conditions such as hydrodynamics, the sterilization process, and increase of the fixation efficiency.

Among the closed bioreactors, there are the bubble columns (BC) and Airlift reactors (ALR), which consist of an inner tube (or riser) where the gas emission is bubbled at its base and an outer tube that surrounds the inner tube (or downcomer), where a flow of liquid or culture medium together with the gas flow forming a circular flow pattern avoiding the need for mechanical agitation, which decreases the energy requirements [13]. It is known that the ALR have shown better mixing, better heat transfer and mass than the BC, due to the presence of the draft tube or riser together with the advantage of simple construction without moving parts [14,15].

It is known that all nutrients in the culture medium, the mass transfer of $CO_2$, the intensity of the light, the gas holdup, and the circulation rate of the liquid in the ALR, have a decisive role in the growth of biomass observed. Therefore, the rate of inlet gas flow leads to gas holdup and the velocity of circulation of the liquid either in the riser or in the downcomer [16,17].

The optimal flow rate of the gas that allows an excellent gas-liquid mixture has great importance in the growth of the microalgae since it will enable to control the pH and the concentration of the $CO_2$ and $O_2$ in the liquid-gas interface [18].

Microalgae and cyanobacteria can consume $CO_2$ for their growth and have the highest efficiency of conversion of $CO_2$ to $O_2$ and products for the synthesis of biomass [19]. The absorption of $CO_2$ by the microalgae depends on the type and quantity of nutrients, the intensity of the light, the time of exposition of light, the temperature, the pH, the $CO_2$ concentration, the gas to liquid flow ratio, the type of reactor, the mass transfer coefficient of $CO_2$, and the microalgae species [20]. It has been reported that the microalgae, in general, can fix $CO_2$ concentrations of up to 20% (*v/v*) [21].

The objective of this work was to propose a two-stage process: The first consists in converting the emissions of a commercial steam generator, such as CO, NO, which are very little soluble in water, into their corresponding $CO_2$ and $NO_2$ oxides utilizing a catalytic converter. The second stage was to use these gases to feed the microalga *Scenedesmus dimorphus*, which grows inside a battery of three airlift bioreactors with an internal loop with different hydrodynamics, light intensity, and light/dark cycles. This study is a proposal for the design of a process using Airlift photobioreactors for the capture of $CO_2$, CO, and NO from emissions of steam generators

## 2. Materials and Methods

### 2.1. Materials

#### 2.1.1. Steam Boiler

In this study, gas emissions from a commercial steam boiler (Cleaver-Brooks) were used (Figure 1). The steam boiler had a steam production of 627 Kg·h⁻¹. The temperature of the gas emission was 150 °C in the chimney (Figure 1k), and its composition was $CO_2$ 14.1 ± 0.21% (*v/v*), $O_2$ 2.3 ± 0.3% (*v/v*), CO 171 ± 5.5 ppm, hydrocarbons 0% (*v/v*), NO 100 ± 0.1 ppm, and $N_2$ 83.9% (*v/v*). The fuel of this boiler was diesel (Figure 1b).

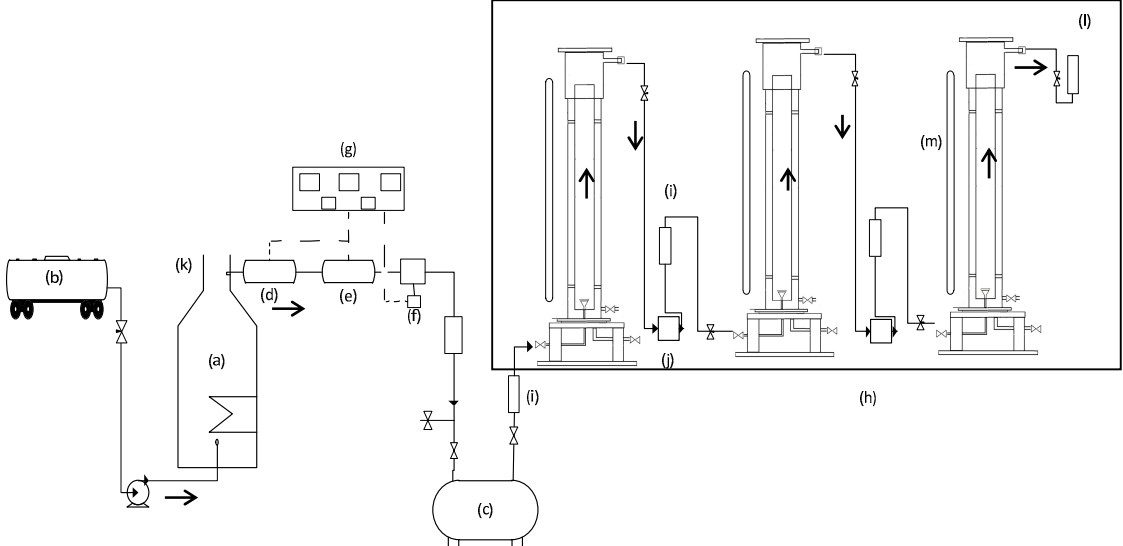

**Figure 1.** Process of $CO_2$ capture from gas emissions. (**a**) Commercial steam boiler. (**b**) Diesel tank. (**c**) Combustion gas storage tank. (**d**) Soot filter. (**e**) Catalytic converter. (**f**) Gas compressor. (**g**) Control panel. (**h**) Biorreactors. (**i**) Rotameters. (**j**) small gas pump. (**k**) boiler chimney. (**l**) Isotermal box and (**m**) Lamps.

### 2.1.2. Catalytic Converter

We used a commercial catalytic converter (Magnaflow 53004) designed to convert the emissions of CO, NO, and hydrocarbons from a combustion engine that burns diesel. This converter had a cell density of 400 cell in$^{-2}$ and was composed of a cordierite ceramic base with a layer of alumina, $CeO_2$, and Pt as the active material to carry out the combustion reactions. The converter had a length of about 10 cm with a diameter of 10 cm.

### 2.1.3. Airlift Bioreactors

In this study, three Airlift bioreactors made of Shott-Duran glass (Germany) with a capacity of 4.271 L each were employed (Figure 1b). These reactors were connected in series to a line of combustion gases coming from the storage tank (Figure 1c). The gas was bubbled into the bioreactors using air diffusers made of sintered glass (average pore diameter 0.16 mm) with a diameter of 3 cm (Figure 1b), which at the same time kept the cultures in constant agitation. We use aquarium air pumps to pass the gas from one reactor to another. The gas flow was controlled by needle valves and rotameters. The bioreactors were placed inside an isothermal acrylic sheet chamber (27 ± 1 °C) with automatic temperature control. The light was provided by fluorescent lamps of 14 W (Figure 1m). The light intensity was measured utilizing a luxometer (Steren, model HER-410). To determine the profiles of the pressure within the riser tube as a function of the length, nine pressure sensors were inserted at distances of 5 cm, which consisted of transparent plastic tubes where the water level was used as a pressure measurement (Figure 2); the pressure values are in Table S1.

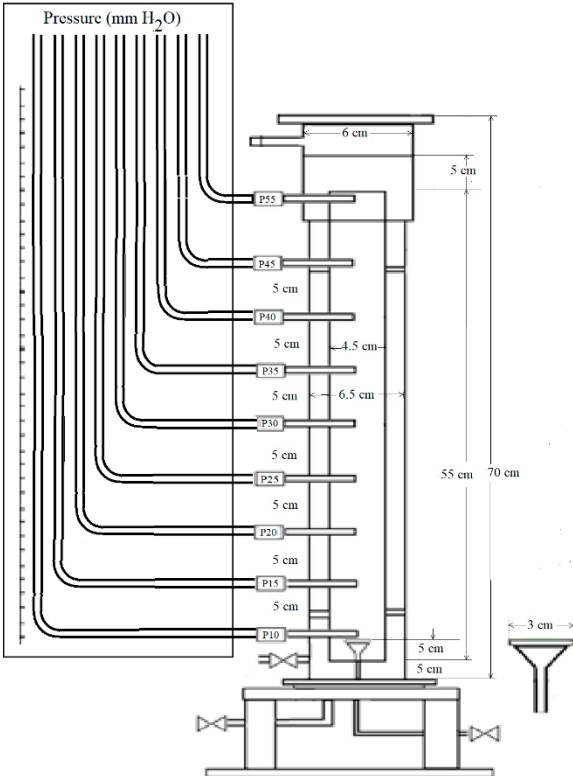

**Figure 2.** Schematic diagram of airlift reactor and its pressure sensors along the riser tube, using a sintering glass disperser.

### 2.1.4. Microorganism and Inoculum

The microalga *Scenedesmus dimorphus* was isolated from a lake located in Mexico City, (coordinates were: 19° 25′ 57.389″ N 99° 7′ 59.549″ W). This microalga was cultured for 12 days at 27 ± 2 °C in culture medium BG-11 [22]. Then, it was centrifuged and suspended in culture medium BG-11 without a carbon source, which was substituted by a gas mixture of air-$CO_2$ of 2% (*v/v*). This gas mixture was fed to the culture for 12 days through bubbling. The culture was carried out in Erlenmeyer flasks with occasional agitation and intensity of light of 17 μmol·m$^{-2}$·s$^{-1}$ with periods of light/dark of 14/10, 16/8 and 24/0 h. The inoculum amount was 20% (*v/v*) of the work volume of each reactor [23].

### 2.1.5. Culture Media

For the cultivation and preservation of the microalga *S. dimorphus*, the BG-11 medium was used in aqueous media and agar slant tubes. The BG-11 medium consisted of (mg·L$^{-1}$): $NaNO_3$, 1500; $MgSO_4 \cdot 7H_2O$, 75; $K_2HPO_4$, 40; $CaCl_2 \cdot 2H_2O$, 36; $Na_2CO_3$, 20; Citric acid, 6; ferric citrate ammonium, 6; disodic EDTA, 1; $H_3BO_3$, 2.86; $MnCl_2 \cdot 4H_2O$, 1.81; $Na_2MoO_4 \cdot 2H_2O$, 0.39; $ZnSO_4 \cdot 7H_2O$, 0.222; $CuSO_4 \cdot 5H_2O$, 0.079; $Co(NO_3)_2 \cdot 6H_2O$, 0.049 [22]. The culture medium was adjusted to pH 7.4, sterilized at 121 °C and 15 lb in$^{-2}$ for 15 min before adding to the bioreactors.

### 2.2. Methods

### 2.2.1. Composition of the Gas Emission

The analyses of $CO_2$, CO, $O_2$, hydrocarbons (HC) and $SO_2$ was carried out in the gas emission of the steam boiler and also at the output of the airlift reactors. The analysis of $CO_2$, CO, and $O_2$ were carried out using a digital gas analyzer (Fyrite® InTech™, Bacharach Model P/N 0024-7341, US). The hydrocarbons were analyzed using a gas chromatograph with FID detector (Varian 3400). The NOx were analyzed using a gas analyzer Testo 340, and finally, the analysis of $SO_2$ was made by



precipitation of the $SO_4^{-2}$ ion with $BaCl_2$ which generates a precipitated $BaSO_4$ that produces turbidity in water, which can be measured spectrophotometrically at a wavelength of 420 nm [24].

### 2.2.2. Procedure to Convert CO, NO, and Hydrocarbons

In this study, the gas flow of the emissions was 88.34 L·min$^{-1}$, and it was compressed in a special tank of 300 L (Figure 1c) at a pressure of 100 psig until the boiler was turned off. The boiler combustion gases passed through a ceramic type filter to remove the soot (Figure 1d), and subsequently, the CO, NO, and Hydrocarbons (HC) were oxidized by a catalytic converter (Figure 1e) designed to oxidize these compounds to $CO_2$ and $NO_2$ at 350 °C by means of a control of electrical resistance (Figure 1g) in order to achieve the highest conversion. These combustion gases were analyzed before and after the catalytic converter.

### 2.2.3. Characterization of the Catalytic Converter by SEM and EDS

To describe this converter, the scanning electron microscopy (SEM) and X-ray energy scattering spectrometry (EDS) were used. The study of the morphology of the catalytic converter was carried out in a microscopy JOEL model JSM 6701F. Samples of the ceramic structure of the catalytic converter were vapored with a solution of Au/Pd to make them conductors; then, these were located on a graphite tape. To carry out the chemical analysis by EDS, a dispersive energy detector was incorporated in the microscope.

### 2.2.4. Cultures

Eight cultures were carried out using this flue gas, where the effects of gas superficial velocity, the intensity of light, light/dark cycles on the growth of microalga *S. dimorphus* were evaluated. The flow rates were 0.05, 0.1, and 0.15 vvm (volume of gas per volume of liquid per minute); the light intensities studied were 27, 60.75, and 76.27 µmol·m$^{-2}$·s$^{-1}$; and the ligth/dark cycles were 14/10, 16/8, and 24/0 h (Table 1).

The cultures were carried out in three bioreactors (Figure 1h) where the combustion gas passed through the catalytic converter. This gas was fed to the reactors continuously and in series so that the gas that left a reactor served as a feed for the next reactor and so on. The temperature was maintained at 27 ± 1 °C, and the pH was not controlled. The cultures were held for 12 days, taking a sample every 24 h. All volumetric flows were controlled by rotameters (Figure 1i), and small gas pumps (Figure 1j) were used to feed each bioreactor.

The temperature of the bioreactors was kept constant using an isothermal chamber (Figure 1g) with acrylic sheets and electrical resistances. The light was provided by a fluorescent lamp of 14Watts (Figure 1m) that was placed in front of each bioreactor.

**Table 1.** Experimental conditions of the cultures.

| Culture | Light Intensity (µmol·m$^{-2}$·s$^{-1}$) | Light/Dark Cycle (h) | Gas Flow (vvm) |
|---------|------------------------------------------|----------------------|----------------|
| C1 | 76.27 | 12/12 | 0.05 |
| C2 | 76.27 | 12/12 | 0.10 |
| C3 | 76.27 | 12/12 | 0.15 |
| C4 | 27 | 14/10 | 0.1 |
| C5 | 60.75 | 14/10 | 0.1 |
| C6 | 76.27 | 14/10 | 0.1 |
| C7 | 60.75 | 16/8 | 0.1 |
| C8 | 60.75 | 24/0 | 0.1 |

2.2.5. Calculation of Gas Holdup ($\varphi_i$)

The gas Holdup is part of the fluid dynamics of airlift reactors and is defined as the volumetric fraction of the gas within the total volume of a gas-liquid dispersion [25].

$$\varphi_i = \frac{V_G}{(V_L + V_G)} \tag{1}$$

where the subindexes *L* and *G* indicate the liquid and gas phase (often the solid phase is minimal), and the subscript *i* indicates the region of the reactor in which the holdup is being considered, which can be in the raiser (*r*), in the downcomer (*d*) or the total reactor (*T*). In our case, the $\varphi_r$ was calculated by the Equation (2) for the riser using the liquid height before ($h_i$) and after the flue gas was injected ($h_f$).

$$\varphi_r = \frac{\left(h_f - h_i\right)}{h_f} \tag{2}$$

2.2.6. Calculation of Superficial Liquid Velocity $U_{lr}$ and $U_{ld}$

The superficial liquid velocity ($Ulr$) can be calculated using the Equation (3), where *Ar* and *Ad* are de cross-sectional areas (m$^2$) of the riser and the downcomer respectively [26].

$$Ulr = 0.66 \left(\frac{Ad}{Ar}\right)^{0.78} (Ugr)^{1/3} \tag{3}$$

*Ugr* is the superficial gas velocity which is calculated with the Equation (4)

$$Ugr = \frac{Q_g}{Ar} \tag{4}$$

$Q_g$ is the volumetric gas flow (m$^3 \cdot$s$^{-1}$). The value of 0.66 is for concentric tube airlift reactors [26]. For its part, the superficial velocity of the liquid in the downcomer *Uld* is

$$Uld = Ulr\left(\frac{Ar}{Ad}\right) \tag{5}$$

The Reynolds numbers for the riser $(Re)_r$ and the downcomer $(Re)_d$ are calculated with Equations (6) and (7):

$$(Re)_r = D_r Ulr \frac{\rho}{\mu} \tag{6}$$

$$(Re)_d = D_d Uld \frac{\rho}{\mu} \tag{7}$$

where $D_r$ is the diameter of the riser, and $D_d$ is the equivalent (annular) diameter of the dawncomer. The bubble rise velocity ($U_b$) can be calculated using the superficial gas velocity ($U_{gr}$) in a multiphase flow which is considered as a possible flow velocity in (m$\cdot$s$^{-1}$) using the Equation (5) [13].

$$U_b = \frac{U_{gr}}{\varphi_i} \tag{8}$$

2.2.7. Calculation of Shear Rate ($\gamma$)

For the determination of the shear rate ($\gamma$) from hydrodynamic principles, different mathematical models have been developed [27,28], in which the rheological properties of the fluid are included or not. In our case, we have calculated the shear rate for superficial gas velocities from 0.0018 m$\cdot$s$^{-1}$ (0.13 L$\cdot$min$^{-1}$) to 0.016 m$\cdot$s$^{-1}$ (1.25 L$\cdot$min$^{-1}$).

In the model proposed by Merchuk and Ben-Zvi [28], the rheology of the culture medium is taken into account and also the injection of power (*P*) that supplies the gas in terms of pressure. The dissipation of energy produced by the bubbles seems a more accurate concept for the calculation of the global shear rate per time unit $\gamma$ ($s^{-2}$).

$$\gamma = \left[ \frac{P_1 \, Ugr \, ln\left(\frac{P_1}{P_2}\right)}{a_b \, L_R^2 \, k} \right]^{1/n} \tag{9}$$

where *k* is the viscosity of the culture ($Pa \cdot s^n$) for Newtonian liquids, $a_b$ is the specific interfacial area ($m^{-1}$), $L_R$ is the aerated reactor height, *n* is the flow behavior index (*n* = 1 for Newtonian fluid), $P_1$ is the pressure at the top of the reactor (atm), $P_2$ is the pressure at the bottom of the reactor (atm). The calculation of interfacial area "$a_b$" can be made using the Equation (10), which takes into account the liquid rheology and the gas superficial velocity [29].

$$a_b = 0.0465 \left( \frac{Ugr^n}{k} \right)^{0.51} \tag{10}$$

2.2.8. Mass Balance of $CO_2$ in the Culture

The evolution of inorganic carbon in the liquid phase ($C_T$), during the algae growth, can be determined by knowing the alkalinity and the pH profiles as a function of time, which are associated with photosynthetic consumption of $CO_2$ and $CO_2$ stripping according to the Equation (11) [30]:

$$d\frac{C_T}{dt} = Kla\left[\left(CO_2^*\right) - (CO_2)\right] - F_C \tag{11}$$

where *KLa* ($h^{-1}$) is the overall volumetric mass transfer coefficient for the $CO_2$, ($CO_2$*) ($mol \cdot L^{-1}$) represents the saturation concentration of $CO_2$ in the liquid phase obtained by the Henry equation, and *Fc* ($mol_{CO2} \cdot L^{-1} \cdot h^{-1}$) represents the $CO_2$ consumption rate by the microalgae photosynthetic activity [22].

$$Fc = P \left( \frac{0.5139}{12} \right) \times 44 \tag{12}$$

$$P = \frac{\Delta X}{\Delta t} \tag{13}$$

where 51.39 wt % is designated as the carbon content of microalgae dry biomass by [19]; 12 ($g \cdot mol^{-1}$) and 44 ($g \cdot mol^{-1}$) represents the molecular weights of carbon and $CO_2$, respectively. The typical molecular formula of microalgal biomass is $CO_{0.48}H_{1.83}N_{0.11}P_{0.01}$ [19]. The composition in weight percent that results is $C_{51.39}O_{32.89}H_{7.83}$, $N_{6.59}P_{1.3}$. *P* ($g_{cel} \cdot L^{-1} \cdot d^{-1}$) is the productivity, where $\Delta X$ is the difference between the maximum biomass *(Xmax)* and the initial biomass (*Xo*) ($g \cdot L^{-1}$), and $\Delta t$ is the culture time (days) [22].

In the case of calculation of ($CO_2$*), using the Henry equation:

$$\left(CO_2^*\right) = \frac{P_{CO2}}{H_{CO2}} \tag{14}$$

$$H_{CO2} = e^{\left[\frac{11.25 - 395.9}{T - 175.9}\right]} \tag{15}$$

where the coefficient $HCO_2$ ($Pa \cdot m^3 \cdot mol^{-1}$) is only in function of the temperature of the culture in Kelvin degrees (*K*) [31].



### 2.3. Determination of Biomass by Optical Density

The microalga biomass was measured by optical density with a spectrophotometer (Spectronic Inst. model 21D) at 678 nm. The readings obtained were interpolated on a calibration curve relating the optical density with the dry weight of *S. dimorphus*. The dry weight of the microalga was measured by filtering 10 mL aliquots through Whatman filter paper No. CF/A (1.6 μm pore size). Each filter plus the microalgae sample were dried at 50 °C to constant weight because it was observed that at higher temperatures the microalga degraded showing a change of color to dark brown. It was found the relationship between optical density and biomass concentration was as follows [10]:

$$y = 0.5847x + 0.0112 \left(R^2\right) = 0.9995 \tag{16}$$

where $y$ refers to the biomass concentration ($g \cdot L^{-1}$), and $x$ refers to the optical density ($OD_{678}$).

### 2.4. Determination of Maximum Specific Growth Rate

The maximum specific growth rate ($d^{-1}$) was calculated using the following equation:

$$\mu max = \frac{ln\left(\frac{X_2}{X_0}\right)}{\Delta t} \tag{17}$$

where $X_2$ is the maximum biomass, $X_0$ is the initial biomass ($g \cdot L^{-1}$), and $\Delta t$ is the culture time where the maximum biomass was obtained (days) [22].

### 2.5. Determination of CO₂ Removal Efficiency

To determine de $CO_2$ removal efficiency, the $CO_2$ concentration was measured on the outlet gas stream of each reactor by a $CO_2$ analyzer (Fyrite® InTech™, Bacharach Model P/N 0024-7341, US). The amount of $CO_2$ removed was calculated using the following equation [20]:

$$CO_2 \text{ removal efficiency } (\%) = \left(1 - \frac{CO_2 \text{ ouput}}{CO_2 \text{ input}}\right) \times 100 \tag{18}$$

where the $CO_2$ output and input were the $CO_2$ concentration on the gas stream of each reactor.

### 2.6. pH, Alkalinity, and Concentration of Dissolved Inorganic Carbon ($C_T$)

The concentration of the alkalinity (alk) was determined by titration with HCl [24], and the pH was measured using a digital pH meter (Conductronic 20), and these determinations were used to calculate the concentration of dissolved inorganic carbon ($C_T$) following the equations used by [32–34]. In this case, it was assumed that, in equilibrium the Equations (20) and (21), all chemical species represented the dissolved total inorganic carbon ($C_T$), and the $CO_2$ concentration in equilibrium was higher than that of $H_2CO_3$.

$$H_2O \leftrightarrow H^+ + OH^- \qquad K_W = [OH^-][H^+] = 10^{-14} \tag{19}$$

$$CO_2 + H_2O \leftrightarrow H_2CO_3 \leftrightarrow HCO_3^- + H^+ \quad K_1 = \frac{[HCO3^-][H^+]}{[CO_2]} = 10^{-6.381} \tag{20}$$

$$HCO_3^- \leftrightarrow CO_3^{2-} + H^+ \quad K_2 = \frac{[CO_3^{2-}][H^+]}{[HCO_3^-]} = 10^{-10.377} \tag{21}$$

The total concentration of inorganic carbon is given by:

$$[C_T] = [CO_2] + [HCO_3^-] + [CO_3^{2-}] \tag{22}$$

The alkalinity for the culture is the sum of the anions present minus the concentration of the protons $[H^+]$.

$$[Alk] = \left[HCO_3^-\right] + 2\left[CO_3^{2-}\right] + [OH^-] - \left[H^+\right] \tag{23}$$

The total inorganic carbon in the culture can be obtained as a function of the alkalinity, the pH, and the ionization fractions ($\alpha_0$, $\alpha_1$ and $\alpha_2$) (Equation (24)). The concentration of $[OH^-]$ is determined by calculating the concentration of $[H^+]$ in accordance with the Equations (19) and (25).

$$C_T = \frac{[Alk] - [OH^-] + [H^+]}{(a_1 + 2a_2)} \tag{24}$$

$$[H^+] = 1 \times 10^{-\text{pH}} \qquad [OH^-] = \frac{\left(1 \times 10^{-14}\right)}{1 \times 10^{-pH}} \tag{25}$$

The ionization fractions ($\alpha_0$, $\alpha_1$ and $\alpha_2$) of each species at equilibrium can be obtained using the pH and the ionization constants $K_1$ and $K_2$ in accordance with:

$$a_0 = \frac{1}{1 + \frac{K_1}{[H^+]} + \frac{K_1 K_2}{[H^+]^2}} \tag{26}$$

$$a_1 = \frac{1}{1 + \frac{[H^+]^2}{K_1 K_2} + \frac{[H^+]}{K_2}} \tag{27}$$

$$a_2 = \frac{1}{1 + \frac{[H^+]^2}{K_1 K_2} + \frac{[H^+]}{K_2}} \tag{28}$$

The concentrations of $[CO_2]$, $[HCO_3^-]$, and $[CO_3^{2-}]$ can be obtained using the Equations (29) to (32).

$$Alk = [HCO_3^-] + 2[CO_3^{2-}] = \alpha_1 \, (C_T) + 2\alpha_2 \, (C_T) \tag{29}$$

The $[OH^-]$ and $[H^+]$ concentrations were not considered in Equation (29) because both numerical values are negligible compared to the carbonate species concentrations.

$$[CO_2] = \alpha_0 \, (C_T) \tag{30}$$

$$[HCO_3^-] = \alpha_1 \, (C_T) \tag{31}$$

$$[CO_3^{2-}] = \alpha_2 \, (C_T) \tag{32}$$

### 2.7. Determination of the Overall Volumetric Mass Transfer Coefficient for $CO_2$ (KLa)

The values of *KLa* were obtained by integration of the Equation (11), and because the *Fc* depends on the biomass concentration, the mean value of *KLa* for each culture was calculated. Also, the *KLa* was determined using the dynamic method, which considers the static and dynamic properties of the liquid, such as the viscosity and the liquid superficial velocity, and is given by Equation (33) [35]:

$$Kla = kU_{gr}^{\alpha}\mu_{eff}^{\beta}\left(1 + \frac{A_d}{A_r}\right)^{\delta} U_{Lr}^{\theta} \tag{33}$$

where $k$, $\alpha$, $\beta$, $\delta$, and $\theta$ are empirical constants. $\mu_{\text{eff}}$ is the viscosity of the liquid (Pa s), and *Ad* and *Ar* are the cross sectional area (m$^2$) of the downcomer and riser, respectively; and $U_{gr}$ and $U_{Lr}$ are the superficial gas and liquid velocity (m·s$^{-1}$) in the riser.

## 3. Results and Discussion

### 3.1. Performance of the Catalytic Converter

The average composition of the flue gases that produce the steam boiler is shown as follows: $CO_2$ 14.1 ± 0.21% (*v/v*), $O_2$ 2.3 ± 0.3% (*v/v*), CO 171 ± 5.52 ppm, NO 100 ± 0.1 ppm, HC 0% (*v/v*), $N_2$ 83.9% (*v/v*). This equipment was constantly generating steam, and no significant changes in operation were observed. After carrying out 4.5 h of operation intermittently with periods of operation of 40 min, the concentrations of the gases after passing through the converter were $CO_2$ 14.40% (*v/v*), $O_2$ 2.30% (*v/v*), CO 37 ppm, NO 10.2 ppm, HC 0% (*v/v*), $N_2$ 83.3% (*v/v*). The catalytic converter transformed 78% of the CO to $CO_2$ and 89.8% of NO to $NO_2$ in the flue gas. The chromatographic analysis of the emissions (Varian 3400) showed the presence of small amounts of aldehydes such as acetaldehyde and acrolein (<22 ppm) which were not observed at the output of the converter. In our experiments, the CO conversion of this catalytic converter was 78% when the converter reached a temperature of 170 °C. Carlsson and Skoglundh [36] obtained 100% of CO conversion using a fresh $Pt/Al_2O_3$ catalyst at 175 °C in the form of monolith having 400 cell/in$^2$. The chemical composition of this catalytic converter was 1 wt.%Pt supported on a monolith usually made of cordierite. During the preparation, the washcoat slurry containing the $H_2PtCl_6$, the $Al_2O_3$, and $H_2O$ covered the 75% of the total support (0.12 g coating per cm$^3$ of monolith catalyst).

Other authors [8] also showed the same CO light-off curves (from 120 °C to 175 °C), and the chemical composition included 0.27 wt.%Pt supported on $\gamma$-$Al_2O_3$ and small amounts of $La_2O_3$ and Pd.

The previous authors [36] used concentrations of 0.1% CO with an excess of $O_2$ of 9%, in the absence of other gases. It was observed that in the ignition experiments, the conversion of CO follows the typical light-off curves, which occurs by forming a loop when the catalytic bed is heated and when it is cooled so that two CO conversion profiles are obtained. In the ascent of temperature, the first conversion began at 140 °C and ended at approximately 175 °C for complete conversion.

In the case of our converter, it worked for 3 months, which is why we believe that it showed a good performance, given the inherent deactivation process.

### 3.2. Characterization of the Catalytic Converter

Employing SEM, it was possible to observe in Figure 3a an image of a cross-section of the cordierite monolith with an increase of 25X. In Figure 3b, there was an amplification of 200X, and in Figure 3c we have an increase of 700X. In Figure 3a the well-defined honeycomb ceramic structure was observed; in Figure 3b, the wall of the monolith was observed with a small layer of alumina on its surface; in Figure 3c the alumina with a porous structure was observed. The analysis of the ceramic structure by EDS of the catalytic converter and the average composition of at least three analyses of the studied region are shown in Table 2. The active compounds are Pt, $CeO_2$, and $Al_2O_3$.

### 3.3. Effect of the Hydrodynamics of the Reactor Airlift on the Microalga Growth

The effect of the flow rates of 0.05, 0.1, and 0.15 vvm on algae growth, Reynolds number (Re), superficial gas and liquid velocities (*Ugr* and *Ulr*), gas holdup in the riser ($\varphi_r$), global shear rate ($\gamma$) in the riser, and on the volumetric mass transfer coefficient of $CO_2$ (*KLa*) was evaluated to help to explain these cultures. In these cultures, we observed that the growth rate of biomass was proportionately higher for the culture made with lower flow rate (Figure 4), in the following order.

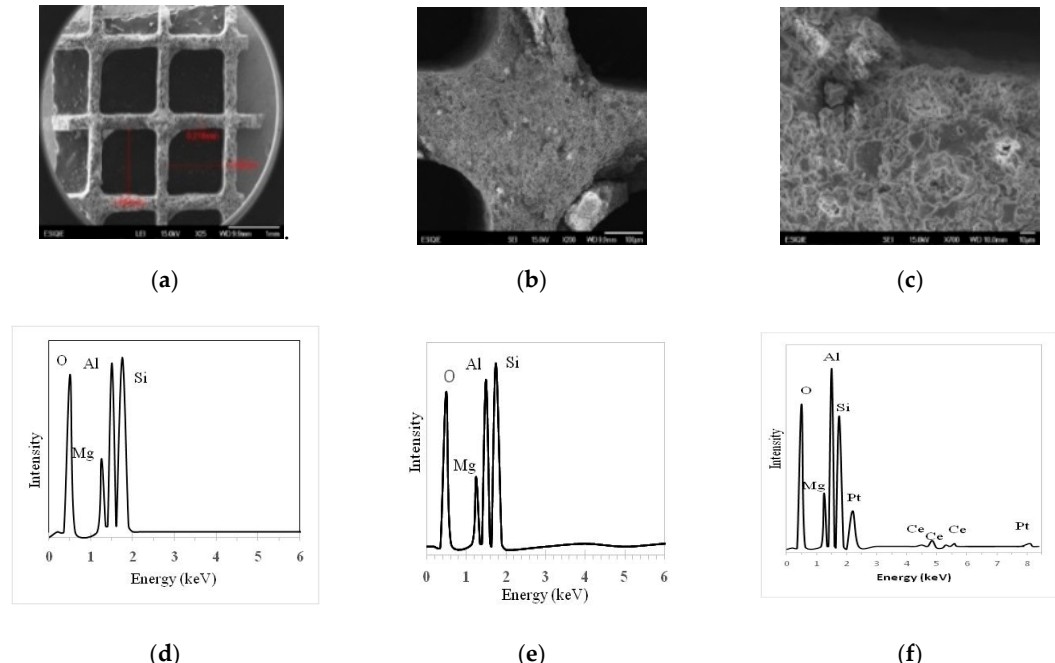

**Figure 3.** Micrograph of the monolith structure observed with magnification of 25X (**a**), 200X (**b**) and 700X (**c**). The corresponding EDS analysis of the monolith structure is shown in (**d**); the analysis on the alumina layer is in (**e**) and the active metals Ce and Pt in (**f**).

**Table 2.** Chemical analysis of the catalytic converter by EDS for the monolithic structure and the active film.

| Chemical Element | In the Monolith Structure (wt.%) | In the Supported Active Phase (wt.%) |
|---|---|---|
| $O_2$ | 50.04 | 41.26 |
| Mg | 7.32 | 5.45 |
| Al | 17.73 | 18.59 |
| Si | 24.91 | 15.18 |
| Ce | 0 | 16.2 |
| Pt | 0 | 3.33 |

C1 > C2 ≈ C3. As shown in Table 3, as the gas flow or the superficial gas velocity in riser (*Ugr*) increases, the velocity of the liquid in the riser (*Ulr*) and the downcomer (*Uld*) increases. In the same way, the gas holdup $\varphi_r$, and the numbers of Re increased from 3121 to 9362 (Table 3), clearly in turbulent regime.

It is clear that a turbulent regime prevails (Re > 2100) within the riser. Even in the downcomer, the turbulent regime is observed at the highest flow (culture C3). With these three experiments (C1 to C3), it is found that the best growth of biomass occurs in the culture C1 and demands flow patterns with numbers of Re less than 6200 and gas holdups less than 0.0082.

The culture C1 showed the maximum biomass of 3.8 g·L$^{-1}$ (Table 4) and the maximum *Fc* (0.53 g $CO_2$ L$^{-1}$·d$^{-1}$). In these cultures, the pH was uncontrolled, but it was observed that it was maintained in a range of 6–7 after the second day of the culture. This behavior could be explained due to the $CO_2$ equilibrium [33] and the metabolism of the microalga itself [37]. Also, it was observed that the exponential growth phase continued beyond day 12.

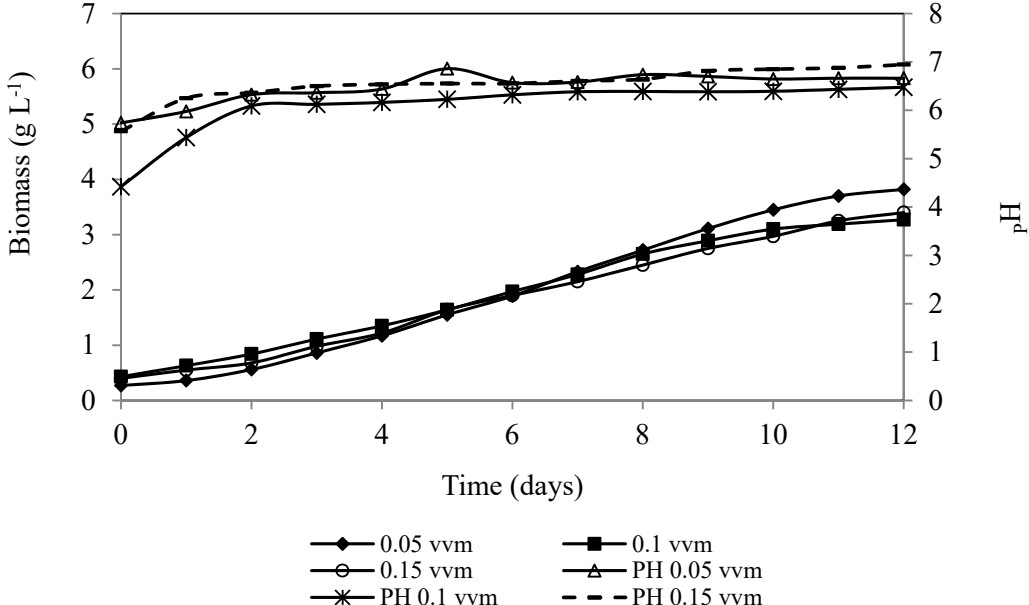

**Figure 4.** Effect of the flow rate on the growth of *S. dimorphus* and the pH.

**Table 3.** Effect of the superficial gas velocity $Ugr$ on the hydrodynamic parameters in each of the airlift bioreactors.

| Culture | $Qg$ (L·min⁻¹) | $Ugr$ (m·s⁻¹) | $Ulr$ (m·s⁻¹) | $Uld$ (m·s⁻¹) | $Ub$ (m·s⁻¹) | $\varphi_r$ | $Re_r$ | $Re_d$ | $(\gamma)$ (1/s) |
|---------|------|------|------|------|------|------|------|------|------|
| C1 | 0.137 | 0.0018 | 0.00078 | 0.00032 | 0.367 | 0.0049 | 3121 | 902 | 8.07 |
| C2 | 0.275 | 0.0036 | 0.0015 | 0.00065 | 0.439 | 0.0082 | 6241 | 1805 | 9.42 |
| C3 | 0.412 | 0.0054 | 0.00234 | 0.00098 | 0.337 | 0.016 | 9362 | 2708 | 10.5 |

This behavior was probably due to better gas exploitation inside the reactor, derived from a lower $CO_2$ stripping caused by the low fed gas flow [38]. Meanwhile, for the cultures C2 and C3 with higher flow rates of 0.1 and 0.15 vvm, the maximum biomasses were of 3.27 and 3.4 g·L⁻¹, respectively (Table 4).

**Table 4.** Productivity $P$, $CO_2$ Fixation rate $Fc$ and $\mu_{max}$ of the cultures.

| Culture | Biomass (g·L⁻¹) | $\mu_{max}$ (d⁻¹) | $P$ (g$_{cel}$ L⁻¹·d⁻¹) | $Fc$ (g$_{CO2}$ L⁻¹·h⁻¹) |
|---------|------|------|------|------|
| C1 | 3.8 | 0.22 | 0.29 | 0.53 |
| C2 | 3.27 | 0.19 | 0.27 | 0.49 |
| C3 | 3.4 | 0.13 | 0.22 | 0.41 |
| C4 | 1.55 | 0.15 | 0.09 | 0.23 |
| C5 | 1.85 | 0.22 | 0.16 | 0.28 |
| C6 | 3.25 | 0.29 | 0.25 | 0.46 |
| C7 | 4.73 | 0.24 | 0.44 | 0.80 |
| C8 | 4.79 | 0.21 | 0.37 | 0.69 |

This decrease is explained because it is known that algae are sensitive to shear rates [25]. It has been mentioned that the Kolmogoroff model of isotropic turbulence [39] indicates that a severe damage can occur at relatively large values of the eddy length scale. This model contemplates a parameter of length or size of the eddy where the energy begins to be dissipated by the viscous resistance.

The increase of superficial gas and liquid velocities ($Ugr$ and $Ulr$) produces an increase in the global shear rate ($\gamma$) observed at the outlet of the gas disperser (Table 3). These values of ($\gamma$) were close to the values reported by Shumpe and Deckwer [29], Zaidi et al. [40], and Nishikawa et al. [41], especially at low superficial gas velocities of 0.0018 and 0.0036 m/s (Table 5); however, at a greater superficial gas velocity (0.0054 m/s) the difference was significant (15%).

**Table 5.** Comparison of global shear rates $\gamma$ in [1/s] from other authors respect the values we obtained in cultures C1, C2, and C3 as a function of *Ugr*.

| Culture | *Ugr* (m·s$^{-1}$) | Shumpe [29] | Zaidi et al. [40] | Nishikawa et al. [41] | Present Work |
|---------|--------------------|-------------|-------------------|-----------------------|--------------|
| C1 | 0.0018 | 6 | 5 | 8 | 8 |
| C2 | 0.0036 | 8 | 8 | 14 | 9.4 |
| C3 | 0.0054 | 12 | 12.5 | 20.8 | 10.5 |

The global shear rate profile against riser length is shown in Figure 5. It can be seen that in the position of the bubbler there are the highest shear rate (at 10 cm height of the bioreactor) and as the flow of bubbles increases, the shear rate decreases until reaching an average value at a height of 20 cm along the riser tube.

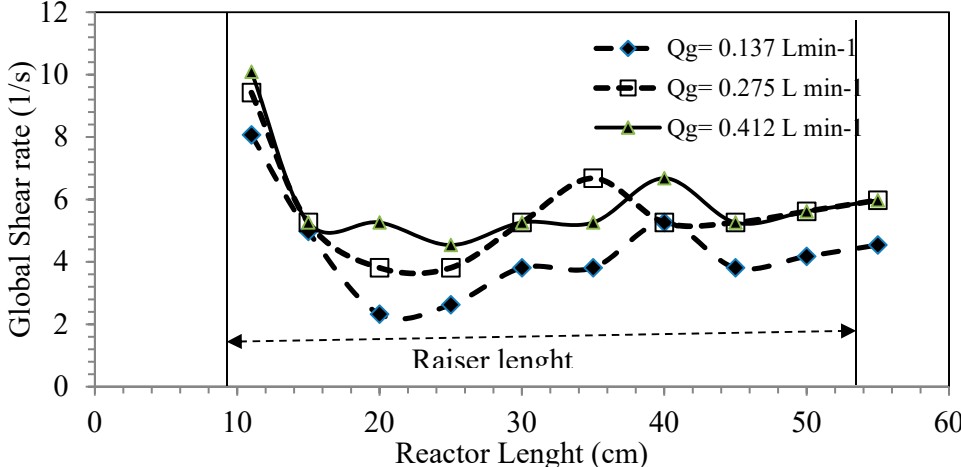

**Figure 5.** Global shear rate in the riser tube in function of the length of the reactor for the three gas flows studied.

As is logical, the global shear rate ($\gamma$) increases with the increase of the superficial gas velocity for the cultures in the following order: C3 > C2 > C1. This same behavior is observed with the volumetric mass transfer coefficients (*KLa*) for $CO_2$ and $O_2$ (Figure 6). This means that at higher superficial gas velocities a higher mass transfer of $CO_2$ could be expected.

Pirouzi et al., 2014 [42], using an Airlift reactor with external loop, obtained values of *KLa* ($O_2$) of 9 to 26 h$^{-1}$ at different superficial gas velocities. Also, Loubiere et al., 2011 [43] used an external Airlift loop reactor, cultivating *Chlamydomonas reinhardtii* obtained a *KLa* of 14.4 h$^{-1}$.

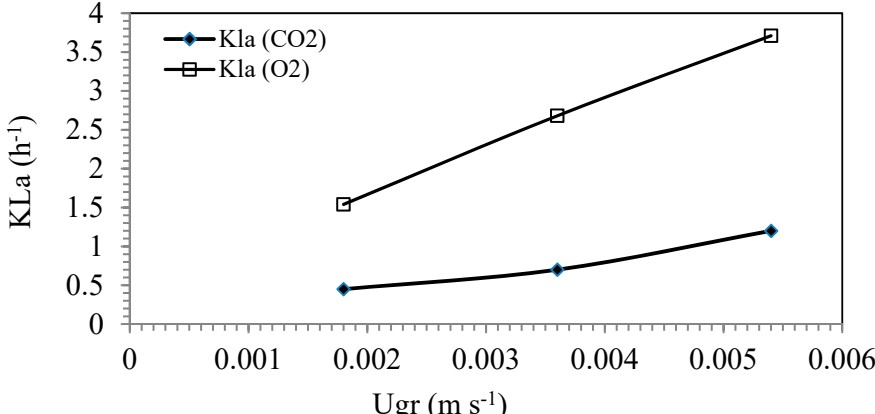

**Figure 6.** Effect of the superficial gas velocity *Ugr* on the volumetric mass transfer coefficient of $CO_2$ (*KLa*$_{CO2}$) calculated by the balance of $CO_2$ and *KLa* ($O_2$) using the dynamic method.

Reyna-Velarde et al., 2010 [44], employing a flat panel Airlift reactor with *Spirulina* sp., obtained a *KLa* of 31.27 $h^{-1}$. Contreras et al., 1998 [45], using a concentric tube Airlift reactor with a culture of *Phaeodactylum tricornutum* obtained a *KLa* of 28.8 $h^{-1}$ for a value of superficial gas velocity of 0.0018 m·s$^{-1}$ and biomass of 4 g·L$^{-1}$. These values of *KLa* ($O_2$) were higher than our results (1.54 to 3.71 $h^{-1}$). This was due mainly to the *Ad/Ar* relationship, since the values of *Ad/Ar* that they report (*Ad/Ar* = 1) were lower than that obtained with the geometry of our reactor (*Ad/Ar* = 2.38), which led to us getting lower values of *KLa*.

However, Kazbar et al., 2019 [46], cultivated *Chlorella* sp. in a flat Airlift reactor in continuous culture and determined the *KLa* experimentally by a de-oxigenation/re-oxigenation method [47] that consisted of removing dissolved oxygen from the liquid phase by injecting gaseous nitrogen ($N_2$) and then monitoring the increase of $O_2$ when switching back to air injection; their value of *KLa* was 1.9 $h^{-1}$. These values are shown in Table 6.

**Table 6.** Comparison of volumetric mass coefficients among different Airlift configurations.

| Airlift Reactor | *KLa* ($h^{-1}$) | Ref. |
|---|---|---|
| External loop (inclined) | 9 to 26 | [42] |
| External loop | 14.4 | [43] |
| Flat panel | 31.27 | [44] |
| Concentric tube | 28.8 | [45] |
| Concentric tube | 1.54 to 3.71 | Present work |
| Flat panel | 1.9 | [46] |

### 3.4. Effect of the Light Intensity on the Growth of Scenedesmus dimorphus

In these experiments, the cultures C4, C5, and C6 with intensities of light of 27, 60.75, and 76.27 µmol·m$^{-2}$· s$^{-1}$ respectively were evaluated. These intensities were used with a constant light/dark cycle of 14/10 h. In the culture C4 at a light intensity of 27 µmol·m$^{-2}$·s$^{-1}$, it was observed that there was a low growth of biomass, reaching a maximum biomass value of 1.55 g·L$^{-1}$ (Table 4). It was also noted that the exponential growth phase was low and extended until 12 days of cultivation (Figure 7). The Productivity and *Fc* were 0.09 g$_{cel}$ ·L$^{-1}$·d$^{-1}$ and 0.23 g$_{CO2}$·L$^{-1}$·d$^{-1}$, respectively (Table 4). This behavior was probably due to the low intensity of light to which the microalga was exposed, so that, the growth was photo-limited [48].

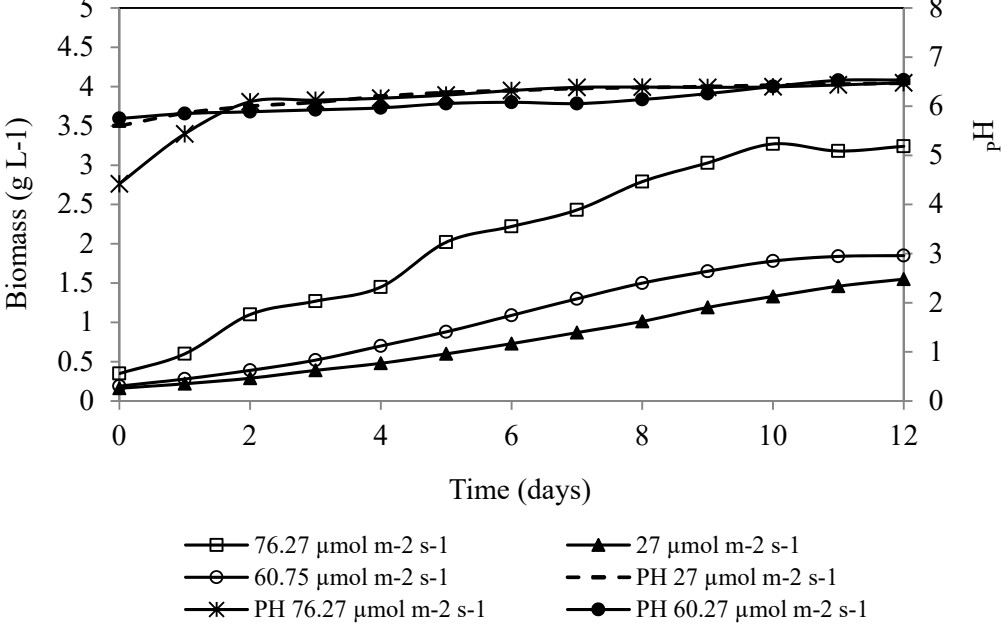

**Figure 7.** Effect of the light intensity on growth of *S. dimorphus.*

In the culture C5 at a light intensity of 60.75 $\mu mol \cdot m^{-2} \cdot s^{-1}$, the microalga had a better growth compared to the culture carried out at 27 $\mu mol \cdot m^{-2} \cdot s^{-1}$, reaching a maximum biomass value of 1.85 $g \cdot L^{-1}$ (Table 4). Besides, the exponential phase was reduced to 9 days (Figure 7), the Productivity and *Fc* were 0.16 $g \, cel \cdot L^{-1} \cdot d^{-1}$ and 0.28 $g_{CO2} \cdot L^{-1} \cdot d^{-1}$, respectively (Table 4). In the culture C6, the light intensity was increased to 76.27 $\mu mol \cdot m^{-2} \cdot s^{-1}$, and *S. dimorphus* reached a biomass of 3.25 $g \cdot L^{-1}$; the productivity and *Fc* were 0.25 $g \, cel \cdot L^{-1} \cdot d^{-1}$ and 0.46 $g_{CO2} \cdot L^{-1} \cdot d^{-1}$, respectively. Jiang et al., 2013, [22] cultured *S. dimorphus* using a simulated flue gas that had 15% (*v/v*) of $CO_2$ with a light intensity of 100 $\mu mol \cdot m^{-2} \cdot s^{-1}$. They obtained a maximum biomass of 2.77 $g \cdot L^{-1}$; the Productivity and *Fc* were 0.362 $g \, cel \cdot L^{-1} \cdot d^{-1}$ and 0.664 $g \, CO_2 \cdot L^{-1} \cdot d^{-1}$, respectively.

### 3.5. Effect of the Light/Darkness Cycle on the Growth of S. dimorphus

In these cultures, the light intensity of 60.75 $\mu mol \cdot m^{-2} \cdot s^{-1}$ remained constant, and the light/dark cycles of 14/10 (culture C6), 16/8 (culture C7), and 24/0 h (culture C8) on the growth of the microalga were evaluated.

It was observed that the biomass had a remarkable increment in the culture C7 with a cycle of 16/8 h, since the maximum value was of 4.73 $g \cdot L^{-1}$ (Table 4 and Figure 8) compared to the culture with the light/dark cycle 14/10 h (C6), which had a maximum biomass of 1.85 $g \cdot L^{-1}$ (Table 4). The productivity and the *Fc* in the culture C7 were 0.44 $g \, cel \cdot L^{-1} \cdot d^{-1}$ and 0.8 $g \, CO_2 \cdot L^{-1} \cdot d^{-1}$, respectively. In the culture C8 where the light was kept on all the time (cycle 24/0), the production of biomass was very close to the biomass of the culture C7 (4.79 $g \cdot L^{-1}$, Table 4), but the specific growth rate was lower (0.21 $d^{-1}$, Table 4).

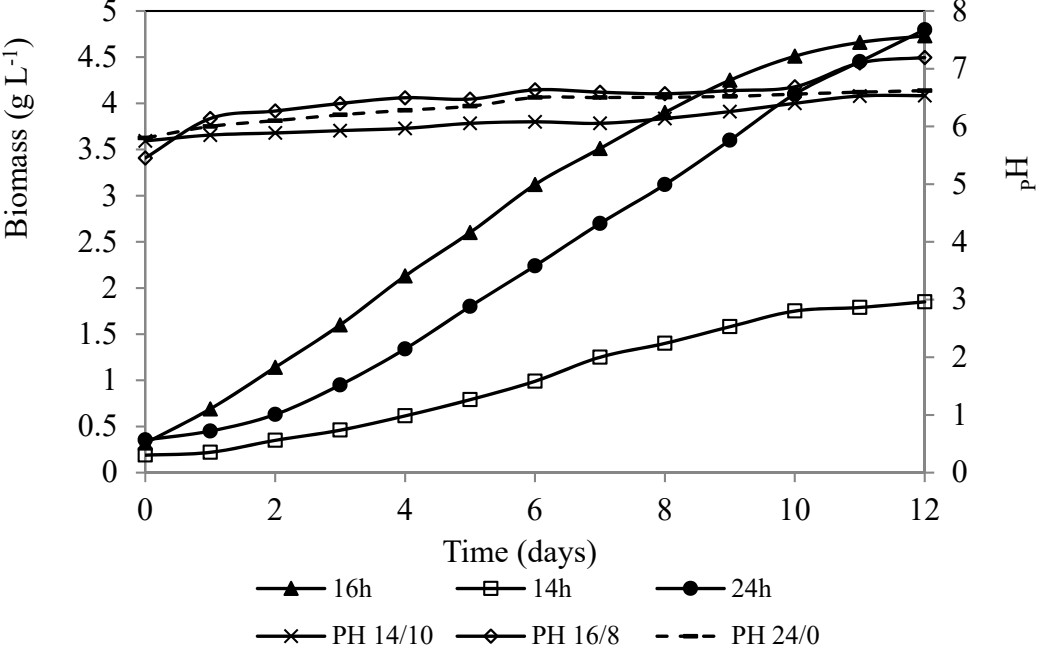

**Figure 8.** Effect of the light/dark cycles on growth of *S. dimorphus*.

These results indicate that light of moderate intensity with a long period of time could increase the growth of biomass. Basu et al., 2013, [49] exposed the microalga *S. obliquus* SA1 to a light intensity of 74.2 $\mu mol \cdot m^{-2} \cdot s^{-1}$, with a light/dark cycle of 14/10 h that was fed with 13.8% $CO_2$. They obtained 4.86 $g \cdot L^{-1}$ of biomass at 34 days of culture. Also, their *Fc* was of 0.225 $g \, CO_2 \cdot L^{-1} \cdot d^{-1}$ which was lower than our results (0.8 $g \, CO_2 \cdot L^{-1} \cdot d^{-1}$).

Thawechai et al., 2016 [50], conducted a study in which *Nannocloropsis* sp. was exposed to several light intensities combined with light/dark cycles and observed that at light intensities of 80 and 120 $\mu mol \cdot m^{-2} \cdot s^{-1}$ with light/dark cycle of 16/8 h, the biomass reached its maximum values, but when

they increased the period to 24 h of continuous light, the biomass did not show any growth. This result was due to the fact that a high intensity of light was coupled with a prolonged period of light, leading to an interruption of growth and subsequent cell death.

### 3.6. Determination of $CO_2$ Removal Efficiency

In this study, the $CO_2$ removal efficiency was determined for all the eight cultures. It can be observed that the $CO_2$ removal efficiency was higher for the cultures with the light intensity of 60.75 $\mu mol \cdot m^{-2} \cdot s^{-1}$ and the light/dark cycles of 16/8 and 24/0, where the maximum value was for the culture C7 (64.3%, Figure 9). Jiang et al., 2013 [22], cultured *S. dimorphus* at 100 $\mu mol \cdot m^{-2} \cdot s^{-1}$ of intensity of light, and their $CO_2$ removal efficiency was of 1.6%, which was lower than our results.

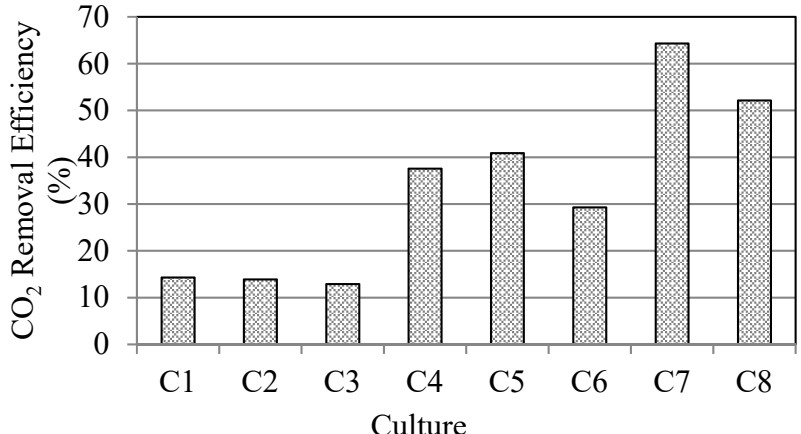

**Figure 9.** $CO_2$ removal efficiency of all cultures.

### 4. Conclusions

The effects of shear rate, light intensity, and light/dark cycles on the biomass growth of the *S. dimorphus* algae were studied. It was observed that low shear rates (Re ≈ 3200) contribute to having a greater amount of biomass. The geometry of the reactor and the *Ad/Ar* relationship had an important influence on the value of the overall mass transfer coefficient (*KLa*) since values of *Ad/Ar* >1 can decrease this parameter. It was observed that the best light intensity and light/dark period were 60.75 $\mu mol \cdot m^{-2} \cdot s^{-1}$ and 16/8 h, respectively. Finally, the catalytic converter used in this study converted the 78% of CO to $CO_2$ and 89.8% of NO to $NO_2$ in the flue gas. This fact helped the converted $CO_2$ and $NO_2$ gases to be assimilated by the microalgae.

**Supplementary Materials:** The following are available online at http://www.mdpi.com/2076-3417/9/16/3212/s1, Table S1: Pressure along the length of riser (in Pa) for the three flow rates.

**Author Contributions:** Conceptualization, A.C.A. and C.J.L.; Methodology, A.C.A., C.J.L. and R.C.; Formal Analysis, A.C.A., C.J.L. and B.Z.; Investigation, A.C.A., C.J.L. and Z.B.; Resources. C.J.L.; Data Curation, A.C.A. Writing: A.C.A. and C.J.L.; Review & Editing, C.J.L.; Supervision, C.J.L.; Project Administration, C.J.L.; Funding Acquisition, C.J.L., R.C. and Z.B.

**Funding:** This research received no external funding.

**Acknowledgments:** The authors gratefully acknowledge the scholarship provided by the National Council of Science and Technology (CONACYT-México) and the technical support of the Metropolitan Autonomous University- Azcapotzalco, México. The authors also thank the Instituto Politécnico Nacinal of México for the analytical facilities given to this research.

**Conflicts of Interest:** The authors declare no conflict of interest.

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
