# Peer review of "CO2 Capture of the Gas Emission, Using a Catalytic Converter and Airlift Bioreactors with the Microalga Scenedesmus dimorphus"

_applsci, doi:10.3390/app9163212_

Round 1
Reviewer 1 Report
The authors carried out the CO2 capture from gas emission of a steam boiler using three air-lift bioreactors in series with a catalyst unit. The study is interesting but it needs to modified. Following questions need to be addressed prior to publication.
General comments:
My mind concern about this study is the lack of comparison of the results obtained. Authors have done a discussion of the results but it were not compared with the bibliographic. For instances:
- P9.L265. Catalytic efficiency. Results about similar catalytic systems used in other studies should be include in order to know if the catalytic performance was good or not.
o P10.Table 1. Chemical composition
o P12. Table3. Shear values
o P10.Table 2. Specific growth rate and biomass productivity
o L310. Values of maximum biomass concentration and maximum Fc.
o P13.L329-335. I believe that the mass transfer coefficient are quite low. For instance, Pirouzi et al. (2014) measured values between 9-26 h-1. Why was not used the dynamic method to obtain the KLa?
Pirouzi, A., Nosrati, M., Shojaosadati, S., Shakhesi, S., 2014. Improvement of mixing time, mass transfer, and power consumption in an external loop airlift photobioreactor for microalgae cultures. Biochem Eng J 87, 25–32. https://doi.org/10.1016/j.bej.2014.03.012
o The intensity of light that produce inhibition (87.75 μmol m-2 s-1) also shocked me. For instance, Nalini and Vijayaraghavan (2018) growth Scenedesmus dimorphus at 110 μmol m-2 s-1 without inhibition and Abu et al. (2017) even at higher values.
Nalini, S. P. K., & Vijayaraghavan, K. (2018). Effect of nitrogen limitation on lipids accumulation in Scenedesmus dimorphus. Indian Journal of Environmental Protection, 38(10), 817-826.
Abu Hajar, H.A., Riefler, R.G. & Stuart, B.J. Bioprocess Biosyst Eng (2017) 40: 1197. https://doi.org/10.1007/s00449-017-1780-4
- It is unknow the CO2 removal efficiency, please add it.
Some more specific comments
- P1.L15. CO converted to CO2, please add it.
- P1.L12-20. Maximum CO2 fixation and biomass productivity should be addressed in the abstract.
- P2.L79-80. This sentence is very ambitious and I believe that is was not reached by the work, please modified it.
- Figure 1. Check the arrows.
- P3.L95. Add the commercial name of the catalyst and the supplier company.
- P3.L111. Pressure along the length was measured but results were not included. Moreover, it can be interesting to know the brand and kind of sensor.
- P4.L129/P5.L164. was the pH controlled in the bioreactor?. If no, what were the pH values in the system?. If the pH was controlled, how much NaOH was consumed?
- P5.L156-160. A new table with number of experiments and experimental conditions should be included.
- P6.L194-197. Check the size of text.
- P7. Equation 11. Fc is not a constant value because it depends of the biomass concentration, How was Fc calculated?. I recommend to solve the systems using Matlab® or Octave® (ode functions).
- P7.L222. The standard methods recommend to dried the dish until constant weight in an oven at 103-105ºC (2540B. Total solid dried method), why was 50ºC used?.
- P7.L226. Add a reference for the alkalinity method.
- P8.Equation 24. [OH-] and [H+] concentration (equation 18) were not consider in equation 24. I suppose because both are negligible small compared to the carbonate species concentrations, please clarified it.
- P8.L242. equation 11?
- P9-P10. Figure 2. Check the position.
- P10.Table 2. Units of the specific growth rate are 1/time (d-1 or h-1).
- P11.Figure 3a. Why were the initial biomass concentrations different?, experiments should be done at the same initial biomass concentration.
- P11. Figure 3b. If three gas flow rates (Table 2) were test, how it is possible to have 10 point in this figure?
- P12 Figure 3c. Lines moved.
Reviewer 2 Report
Manuscript Number: Applesci-529479-peer-review-v1
Title: An integrated process to CO2 capture from a gas emission of a steam boiler, utilizing a catalytic converter and Airlift bioreactors with the microalga Scenedesmus dimorphus.
Article Type: Research Paper
General comments:
Overall manuscript is well written, and this manuscript has scientific merits. I just concern about the title. Title is too big. An ideal title should contain max 20 words. I suggest this manuscript for minor revision.
Round 2
Reviewer 1 Report
Congratulations for the good work, answers and changes made.